# Near Real-Time Monitoring of the Christmas 2018 Etna Eruption Using SEVIRI and Products Validation

**Stefano Corradini [1,\*], Lorenzo Guerrieri [1], Dario Stelitano [1], Giuseppe Salerno [2], Simona Scollo [2], Luca Merucci [1], Michele Prestifilippo [2], Massimo Musacchio [1], Malvina Silvestri [1], Valerio Lombardo [1] and Tommaso Caltabiano [2]**

[1]  Istituto Nazionale di Geofisica e Vulcanologia (INGV), ONT, 00143 Rome, Italy;
    lorenzo.guerrieri@ingv.it (L.G.); dario.stelitano@ingv.it (D.S.); luca.merucci@ingv.it (L.M.);
    massimo.musacchio@ingv.it (M.M.); malvina.silvestri@ingv.it (M.S.); valerio.lombardo@ingv.it (V.L.)

[2]  Istituto Nazionale di Geofisica e Vulcanologia (INGV), OE, 95123 Catania, Italy;
    giuseppe.salerno@ingv.it (G.S.); simona.scollo@ingv.it (S.S.); michele.prestifilippo@ingv.it (M.P.);
    tommaso.caltabiano@ingv.it (T.C.)

\*   Correspondence: stefano.corradini@ingv.it; Tel.: +39-06-51860621

**Abstract:** On the morning of 24 December 2018, an eruptive event occurred at Etna, which was followed the next day by a strong sequence of shallow earthquakes. The eruptive episode lasted until 30 December, ranging from moderate strombolian to lava fountain activity coupled with vigorous ash/gas emissions and a lava flow effusion toward the eastern volcano flank of Valle del Bove. In this work, the data collected from the Spinning Enhanced Visible and InfraRed Imager (SEVIRI) instruments on board the Meteosat Second Generation (MSG) geostationary satellite are used to characterize the Etna activity by estimating the proximal and distal eruption parameters in near real time. The inversion of data indicates the onset of eruption on 24 December at 11:15 UTC, a maximum Time Average Discharge Rate (TADR) of 8.3 $m^3$/s, a cumulative lava volume emitted of 0.5 $Mm^3$, and a Volcanic Plume Top Height (VPTH) that reached a maximum altitude of 8 km above sea level (asl). The volcanic cloud ash and $SO_2$ result totally collocated, with an ash amount generally lower than $SO_2$ except on 24 December during the climax phase. A total amount of about 100 and 35 kt of $SO_2$ and ash respectively was emitted during the entire eruptive period, while the $SO_2$ fluxes reached peaks of more than 600 kg/s, with a mean value of about 185 kg/s. The SEVIRI VPTH, ash/$SO_2$ masses, and flux time series have been compared with the results obtained from the ground-based visible (VIS) cameras and FLux Automatic MEasurements (FLAME) networks, and the satellite images collected by the MODerate resolution Imaging Spectroradiometer (MODIS) instruments on board the Terra and Aqua- polar satellites. The analysis indicates good agreement between SEVIRI, VIS camera, and MODIS retrievals with VPTH, ash, and $SO_2$ estimations all within measurement errors. The SEVIRI and FLAME $SO_2$ flux retrievals show significant discrepancies due to the presence of volcanic ash and a gap of data on the FLAME network. The results obtained in this study show the ability of geostationary satellite systems to characterize eruptive events from the source to the atmosphere in near real time during the day and night, thus offering a powerful tool to mitigate volcanic risk on both local population and airspace and to give insight on volcanic processes.

**Keywords:** satellite remote sensing; volcanic ash and $SO_2$ retrievals; volcano monitoring; Etna 2018 eruption; ground-based remote sensing; volcanic hazard

## 1. Introduction

Explosive volcanic eruptions impact both society and the environment. The dispersion of volcanic ash in atmosphere represents a threat for aviation safety as it may cause damage to aircraft systems and

engines [1]. The long-term persistence of ash and aerosol sulfate in atmosphere can influence the climate by reflecting solar radiation to space [2]. Moreover, tephra fallout can cause collapses of roofs and infrastructures, and together with gas emission, it may strongly affect human and animal health, vegetation, and infrastructures [3–8]. Real-time monitoring volcanic eruption offers now a powerful tool to mitigate risks on both local population and air traffic and gain insight into processes and mechanism of volcanic unrest. Technological advances of remote sensing systems made over the last 20 years marked a major step forward in both the proximal and distal monitoring of volcanic eruptions. In particular, exploiting their high acquisition time and the use of infrared channels, the geostationary satellite systems offers a unique opportunity to track the whole evolution of the volcanic eruptions, extending the monitoring capabilities during both day and night [9,10].

The volcanic proximal monitoring mainly concerns the detection of the thermal anomalies and the estimation of Eruption Source Parameters (ESPs). ESPs include the height of the eruptive column (or Volcanic Plume Top Height (VPTH)), the Total Grain Size Distribution (TGSD), and the Mass Eruption Rate (MER) [9–14]. While VPTH represents a key input for volcanic cloud retrieval procedures [15–18], all terms are crucial for a correct initialization of the volcanic ash dispersion and deposition models [19–22]. The time evolution of the thermal anomalies gives information of the variation of volcano activity and, in particular, allows the computation of the Eruption Start and Duration (ESD) [10,23,24] and the Time Average Discharge Rate (TADR) [25–30] that, if integrated, gives the amount of the lava volume erupted [31–33]. For distal monitoring, the estimation of the altitude of the volcanic cloud (or Volcanic Cloud Top Height (VCTH)), the ash and gases masses and fluxes are carried out. The precise characterization of all these terms is crucial for the mitigation of the volcanic risk and gives insights into volcanic processes [34–36]. Moreover, the volcanic cloud ash/gases products assimilated into dispersion models significantly improve the volcanic cloud forecast [37–39].

Etna (Sicily, Italy, see Figure 1a) is one of the most active volcanoes in the world. Since 2011, it has showed an intense sequence of explosive activities that spanned from strombolian to short-lasting but astonishing episodes of lava fountains that fed plumes [40–42]. This eruptive regime gradually switched since 2016 to long-lasting mild explosive strombolian activities coupled with isolated episodes of lava flows from the summit crater of the volcano [43]. The Christmas 2018 Etna eruption was preceded by a period of moderate explosive activity and small lava flows at the summit craters [44,45]. On the morning of 24 December, this eruptive style suddenly increased, and a 2 km long eruptive fracture opened on the southeaster flank of the volcano [46], spilling out a lava flow toward the Valle del Bove (see Figure 1b). The eruption produced a vigorous ash column, which spread up to about 8 km above the sea level (asl), dispersing south-eastwards of the volcano and causing a disruption to Catania international airport. The explosive activity that was mainly produced by several vents along the fracture [47] decreased since the late afternoon of 24 December and ended the day after. Lava flow gradually ended on 27 December. Figure 1c,d show the 24 December paroxysm and the 27 December activity seen from the Enna and Taormina towns, respectively.

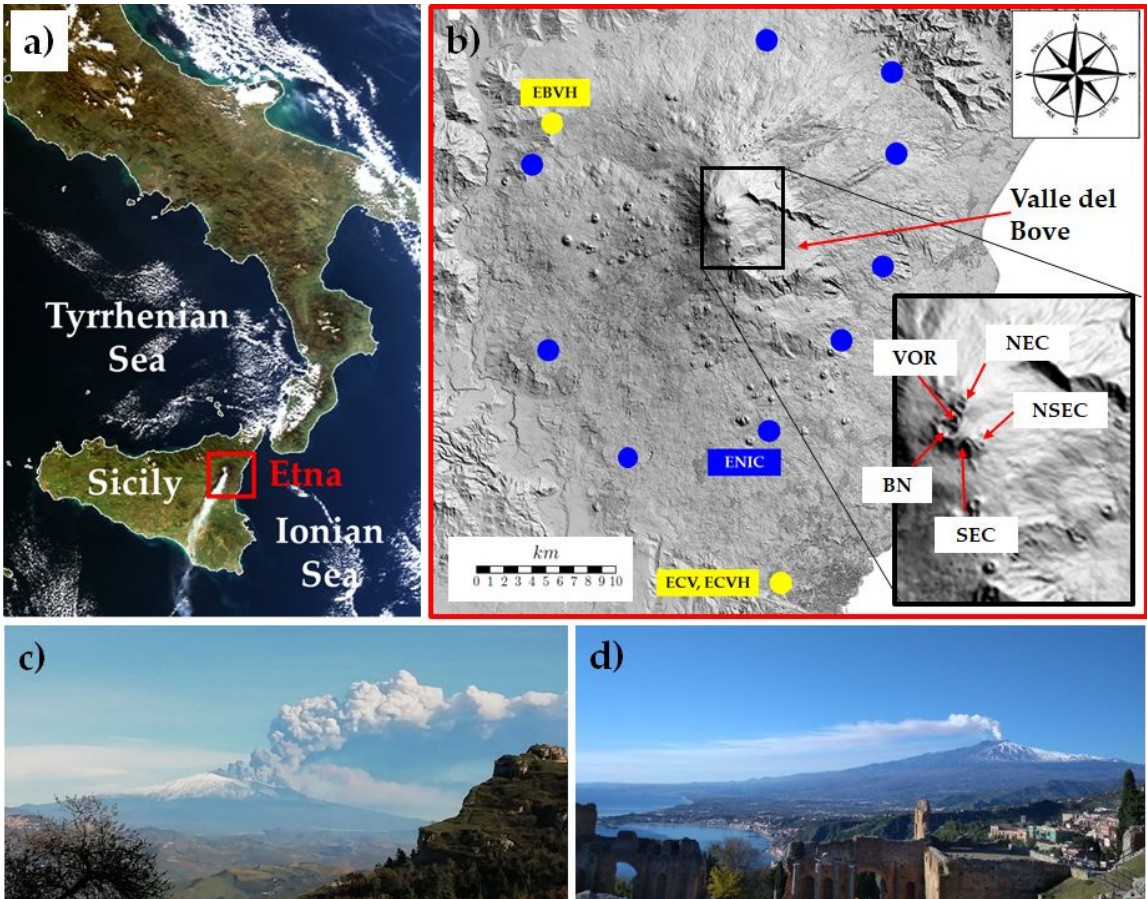

**Figure 1.** (**a**) Image of southern Italy collected from the MODerate resolution Imaging Spectroradiometer (MODIS) satellite sensor on board the Terra/Aqua NASA satellites on 27 December 2018 at 12:20 UTC. The Etna location is identified by the red square. (**b**) Map of Etna volcano with the indication of the visible cameras (yellow points) and FLux Automatic MEasurements (FLAME) spectrometers (blue points) used in this work for the products' validation. In the zoom, the main active craters are indicated: Voragine (VOR), Bocca Nuova (BN), North East Crater (NEC), South East Crater (SEC), and New SEC (NSEC). (**c**,**d**) Etna activities of 24 and 27 December 2018 seen from Enna and Taormina, respectively (photo credits Truscia and Ponzo).

In this work, the data collected by the Spinning Enhanced Visible and InfraRed Imager (SEVIRI) instrument on board Meteosat Second Generation (MSG) geostationary satellite were inverted for the near-real-time proximal and distal characterization of the 24–30 December 2018 Etna eruption. Exploiting its wide on-site monitoring capabilities, Etna allows the unique possibility of validating the satellite retrievals with those obtained from several ground-based instruments.

From the SEVIRI data, the proximal ESD, TADR, lava volume, VPTH, distal VCTH, ash/SO$_2$ masses, and fluxes are computed in near real time. The retrievals are compared with those obtained from the MODerate resolution Imaging Spectroradiometer (MODIS) sensors on board Terra/Aqua polar orbit satellites and the ground-based systems visible (VIS) cameras and Differential Optical Absorption Spectroscopy (DOAS) FLux Automatic MEasurements (FLAME) scanning spectrometers network deployed on Etna. The results show the SEVIRI capability to fully characterize, in space and time, an eruptive event, thus offering a powerful tool to reduce the volcanic risk on local population and airspace and to improve the knowledge of volcanic processes.

The article is organized as follows: Section 2 describes the SEVIRI satellite instrument and the retrieval strategies adopted for the computation of the different proximal and distal parameters. The results are presented and discussed in Section 3, while in Section 4, the comparison with those

obtained from ground-based and satellite-based instruments is realized. Final conclusions are drawn in Section 5.

## 2. SEVIRI Sensor and Retrievals Strategies Description

SEVIRI, on board the MSG geostationary satellite, has 12 spectral channels from visible (VIS) to Thermal InfraRed (TIR), a nadir spatial resolution of 3 km at sub satellite point, and a temporal resolution that ranges from 5 min (rapid scan mode over Europe and Northern Africa) to 15 min (Earth full disk) [48]. From 2008, a satellite acquisition system has been developed at Istituto Nazionale di Geofisica e Vulcanologia (INGV) to collect the SEVIRI images in real time. This former system has been recently upgraded, allowing the real-time acquisition of many other geostationary and polar satellites' datasets. All the SEVIRI images used in this work have been resampled in a regular grid of $3 \times 3$ km$^2$.

The proximal monitoring is realized by estimating ESD, TADR, lava volume, and VPTH by exploiting the SEVIRI channels centred at 3.9 and 10.8 µm (SEVIRI channels 4 and 9 respectively).

ESD is obtained from the MS2RWS (MeteoSat to Rapid Response Web Service) algorithm, based on the difference between the SEVIRI 3.9 µm radiances of the pixel centered on the craters and the mean value from the $5 \times 5$ pixels around it (DR(t)), compared with reference values from historical time series (DT(t)) [9]. DR(t) greater than DT(t) indicate the presence of a thermal anomaly and the possible start of an eruption.

TADR is estimated by exploiting the AVHotRR routine developed by Lombardo [30] based on the knowledge of the SEVIRI 3.9 µm top of atmosphere radiances computed at ambient (Ta) and lava (Tc) temperatures [25–27]. The big uncertainties on Tc causes the higher uncertainty on TADR estimation. Its time integration gives the cumulative lava volume erupted.

VPTH is obtained by applying the consolidated "dark pixel" procedure [49], which is based on the comparison between the minimum SEVIRI 10.8 µm brightness temperature of a pixel contained in a fixed area over the summit craters and the atmospheric temperature profile measured in the same area and at the same time of satellite acquisition [10,11].

For the distal monitoring, VCTH is computed by applying the same procedure described for the VPTH parameter but considering the whole volcanic cloud, while the ash and SO$_2$ detection and retrievals are obtained by exploiting the SEVIRI channels centred at 8.7, 10.8, and 12 µm (SEVIRI channels 7, 9, and 10, respectively). The detection is realized through a Red Green Blue (RGB) composite obtained by combining the brightness temperatures (T$_b$) of the three SEVIRI channels (R: $T_{b,8.7} - T_{b,10.8}$; G: $T_{b,12} - T_{b,10.8}$; B: $T_{b,10.8}$), while the quantitative estimation of the different ash and SO$_2$ parameters are computed by exploiting the Volcanic Plume Retrieval (VPR) procedure, formerly Volcanic Plume Removal [16–18,50]. VPR requires in input only the VCTH and is based on the computation of a new image by replacing the radiance values in the region occupied by the plume with those obtained from a simple linear regression of the radiance outside the edges of the plume itself. The original and the new images allow the estimation of plume transmittances for the TIR bands centred at 8.7, 10.8, and 12 µm from which ash parameters (Aerosol Optical Depth (AOD), effective radius (Re), mass (Ma)) and SO$_2$ (Ms) mass are computed.

The processing of the proximal parameters ESD and TADR is realized by exploiting the SEVIRI images every 5 min (Rapid Scan Mode), while the VPTH and distal parameters (VCTH, ash/SO$_2$ retrievals) are obtained by processing the SEVIRI images every 15 min. The use of the 5-min time rate for the generation of all the products will be implemented in the near future.

Figure 2 shows the SEVIRI ash/SO$_2$ volcanic clouds detection for several images from 24 to 30 December 2018 by applying the described RGB composite. The presence of volcanic ash and SO$_2$ are identified by the red and green colors, respectively.

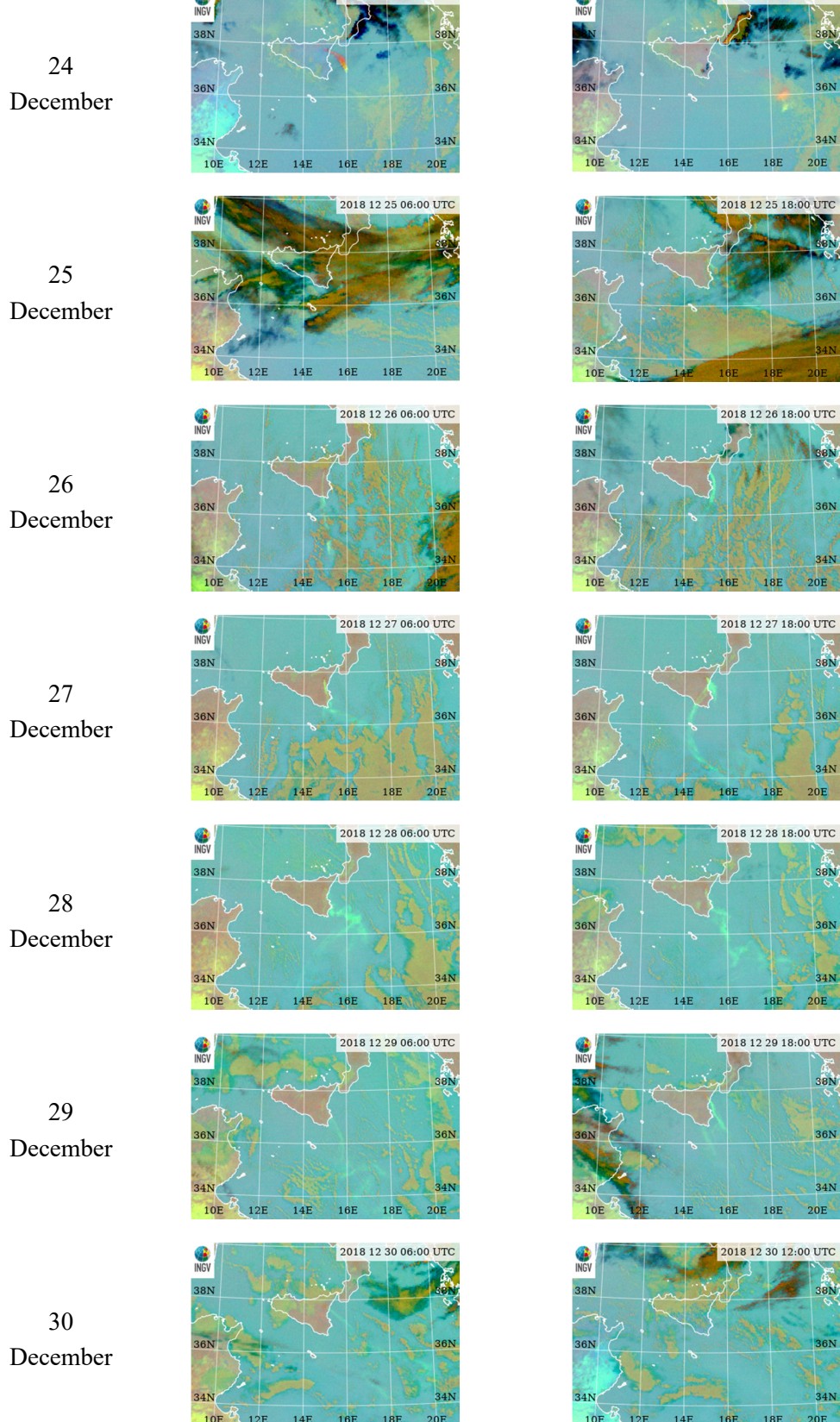

**Figure 2.** RGB composite (R: $T_{b,8.7}-T_{b,10.8}$; G: $T_{b,12}-T_{b,10.8}$; B: $T_{b,10.8}$) showing the Etna volcanic cloud detection considering several Spinning Enhanced Visible and InfraRed Imager (SEVIRI) images collected from 24 to 30 December 2018. The prevalence of red and green colors identify ash and $SO_2$, respectively.

The analysis of the complete time series of the SEVIRI RGB composite images shows that the volcanic cloud flown toward South-South East (S-SE) for the entire period of the eruption. The satellite images show also the persistence of a wide meteorological cloud system over the Mediterranean region from about 00:00 UTC of 25 December to about 08:00 UTC of 26 December, which makes the eruption cloud detection (therefore the retrievals) critical when even impossible. Finally, after 24 December, the volcanic cloud appears mainly green, indicating the prevalence of $SO_2$ than ash.

## 3. Results

In the following paragraphs, the results of the proximal and distal monitoring of the 24–30 December 2018 Etna eruption using SEVIRI data are described. All the times inserted in the paper are in Coordinated Universal Time; then, the suffix "UTC" will be omitted.

### 3.1. Eruption Start and Duration (ESD)

In Figure 3 the orange, blue, and gray lines represent the Dynamic Threshold (DT(t)), the radiance measured at 3.9 μm for the central pixel (L(x*,y*)) and the Difference of Radiance (DR(t)) time series, respectively [9], obtained from the SEVIRI measurements collected every 5 min from 00:00 on 24 December to 23:55 on 30 December. The green area indicates the time interval of the meteorological cloud system present on the area of measurements.

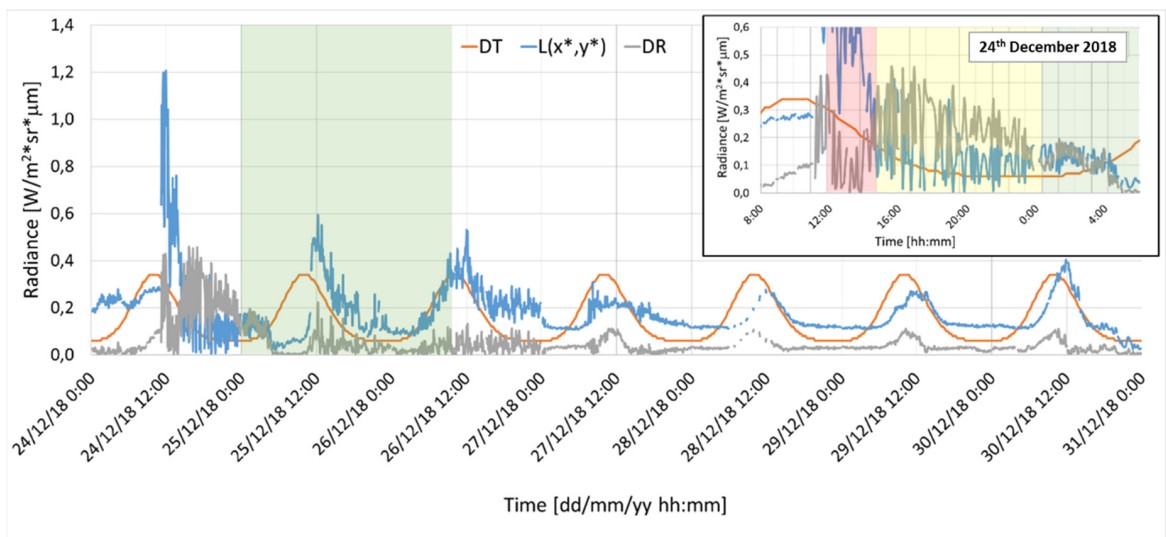

**Figure 3.** Time series of Dynamic Threshold (DT(t), orange line), radiance at 3.9 μm of the pixel centred on Etna summit craters (L(x*,y*), blue line), and Differences of Radiance (DR(t), gray line), obtained from SEVIRI images collected every 5 min. The green rectangle indicates the time interval of the meteorological cloud system over Sicily. The box on the upper right corner is the zoom of the 24 December measurements.

In the upper right corner of Figure 3, a blow-up of 24 December 2018 is displayed. Here, the abrupt increase of DR(t) becomes greater than DT(t) at 11:15, thus identifying the eruption onset. The following decrease of DR(t) from 12:00 to about 15:00 (red area) is caused by the presence of volcanic ash plume over the summit craters that absorb the radiation coming from the surface [10], while the higher values of DR(t) from 15:00 to about 00:00 of 25 December (yellow area) are related to the occurrence of the lava flow from the eruptive fracture. The meteorological clouds presence in the area after 00:00 on 25 December (green areas in zoom and main plots) make the hot spot detection extremely critical. After 08:00 on 26 December, the DR(t) values are always lower than DT(t), indicating the absence of lava flow.

## 3.2. TADR

Figure 4 shows the TADR derived from SEVIRI data from 24 to 27 December by considering the lava temperature ranging from 100 (TADR min, maroon line) to 600 °C (TADR max, blue line) [30]. The maximum derived TADR is 8.3 m³/s at 11:30 on 24 December. The flowing TADR decrease is induced by the presence of the volcanic plume that shielded the radiation coming from the surface (red area). At the end of the paroxysm, TADR shows an exponential decrease (yellow area), until reaching the values prior to the eruption in about one day. As in Figure 3, the green area indicates the presence of the meteorological clouds. By integrating the TADR time series, a maximum cumulative lava volume of 0.5 M m³ is obtained. TADR and total volume retrieved are in good agreement with those obtained by Laiolo et al. [44], which is derived from the analysis of the MODIS sensor on board the Terra/Aqua polar satellites, while significant discrepancies are found with the result obtained in Calvari et al. [45] that uses the HOTSAT system [29,31,32].

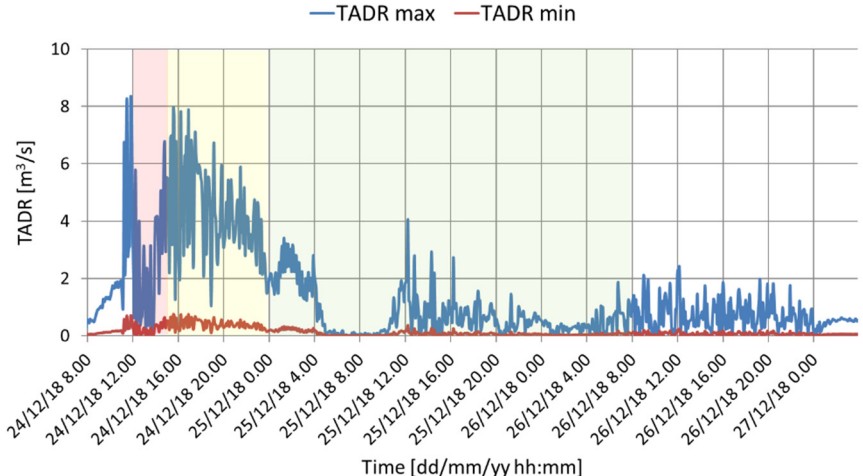

**Figure 4.** Time Averaged Discharge Rate (TADR) time series derived from SEVIRI data from 24 to 27 December 2018. Blue and maroon lines indicate the maximum and minimum TADR values respectively. The red, yellow, and green areas emphasize the influence of volcanic plume, the lava flow, and the presence of meteorological cloud.

Among the TADR computation, the methodology proposed by Lombardo et al. [51], which provided the capability to use high temporal resolution data to predict erupting styles, has been applied. This statistical approach, based on the wavelet transform of time series of SEVIRI radiance data acquired at 5–15 min acquisition rate in the Mid InfraRed (MIR) spectral band (3.9 µm), showed potential in predicting eruptive style. Statistic kurtosis and gradient derived from the analysed time series allowed recognizing the 24 December event as characterized by Strombolian and explosive activity, which are associated with ash emissions. This is in good agreement with field observations and remote sensing measurements.

## 3.3. Volcanic Plume Top Height (VPTH) and Volcanic Cloud Top Height (VCTH)

Figure 5A shows the 10.8 µm brightness temperature for SEVIRI image collected on 24 December 2018 at 12:00. Figure 5B shows the region of 9 × 9 pixels centered on Etna craters (blue square) and the volcanic cloud detection mask (green area) considered for the VPTH and VCTH computation, respectively. Finally, Figure 5C displays the VPTH (blue line) and VCTH (green line) time series obtained by processing the 24 December 2018 SEVIRI images every 15 min. Data were inverted using the atmospheric temperature profiles from the mesoscale model of the hydrometeorological service of Agenzia Regionale per la Protezione Ambientale (ARPA) Emilia Romagna, which is named ARPA-SIM [52], and considering an hourly model output from 72-h weather forecast provided for Etna every 3 h. Before the comparison with the atmospheric temperature profiles, the SEVIRI 10.8 µm

brightness temperatures were decreased by 2 K to take into account the possible non-complete opacity of the pixel [49]. The uncertainties were estimated by considering +/− 2 K of the value obtained from the dark pixel brightness temperature computed [49]. As the plot shows, the estimated maximum height reached by the eruptive column is about 8 km. After the paroxysmal phase, the VPTH started to decrease (eruptive column no longer sustained) and then inverted again its trend after 13:30. Nevertheless, this temporal change is an artefact, since the paroxysmal phase was over. This behavior is due to the formation of a wide orographic cloud over the Etna area that will be discussed in Section 4.1. Instead, VCTH remains approximately constant until 12:45 to gradually decrease until 15:00 (the cloud moved following the wind transport). Note that most likely the VCTH decrease does not represent an actual volcanic cloud height decrease, but it is due to the volcanic cloud dilution. In fact, the dark pixel procedure gives reliable results only when the pixels are completely opaque, but when the volcanic cloud starts to dilute, the pixels become partially transparent, and the procedure fails, underestimating the cloud height.

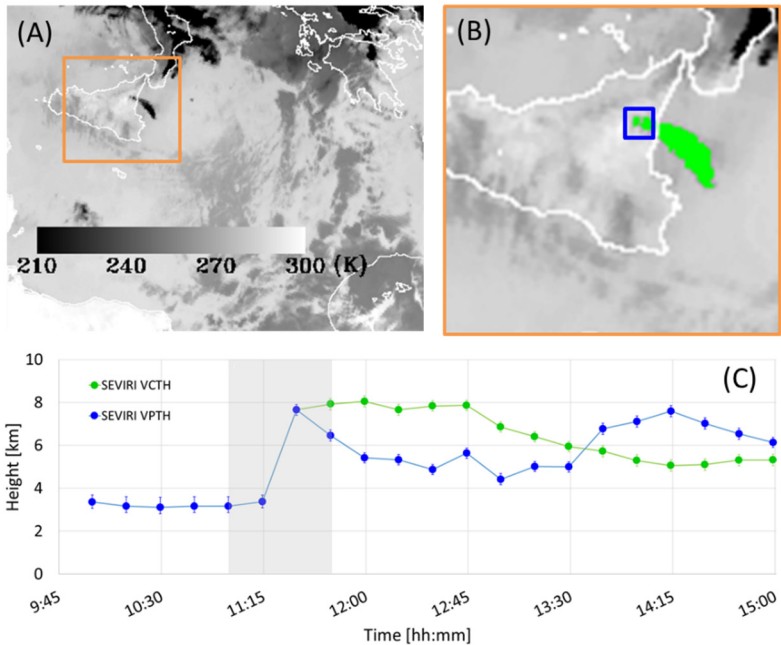

**Figure 5.** (**A**) Brightness temperature at 10.8 µm considering the SEVIRI image collected on 24 December 2018 at 12:00. (**B**) Image of Sicily with the blue square area and the green area used for the Volcanic Plume Top Height (VPTH) and Volcanic Cloud Top Height (VCTH) computation, respectively. (**C**) Plot of VPTH (blue points-line) and VCTH (green points-line) time series obtained from SEVIRI data collected every 15 min on 24 December 2018. The green and blue vertical bars represent the VCTH and VPTH SEVIRI retrieval uncertainties, respectively.

From 25 to 30 December, the volcanic cloud was too transparent even close to the summit craters, making the dark pixel procedure inapplicable. As a result of that, and being the volcanic cloud height required for the ash and $SO_2$ quantitative retrievals based on the VPR procedure, this parameter was evaluated by exploiting the independent measurements of the ground-based calibrated VIS cameras network in Catania (ECV) and Bronte (EBVH) on the west side of Etna at 760 m asl (see yellow circles in Figure 1) [9–11]. Figure 6 shows one camera frame for each day from 26 to 30 December 2018. Note that all the images derive from the ECV camera placed at Catania (CUAD) except for that on 28 December, for which the Bronte (EBVH) VIS camera installed on the western side of Etna was used. During this day, the plume flowed exactly toward the south, making the height retrieval from the ECV camera unfeasible.

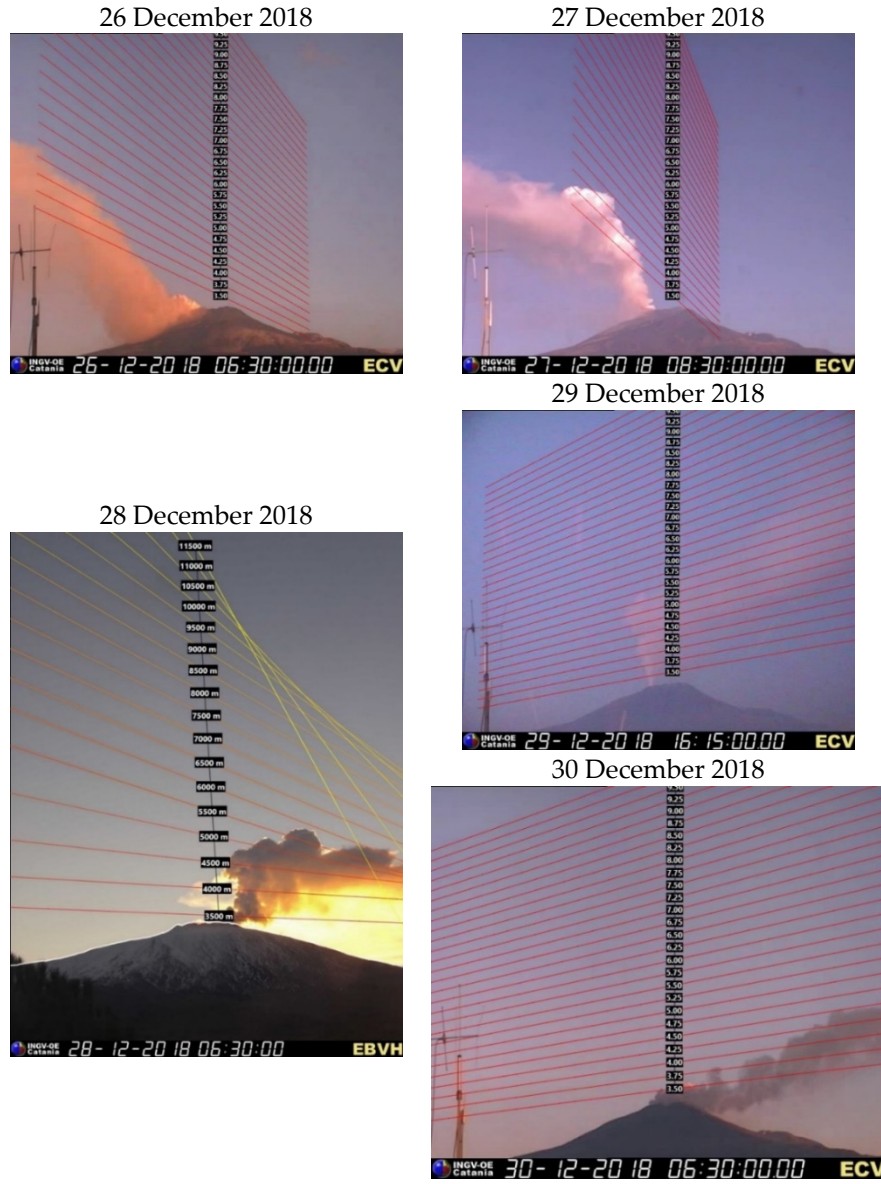

**Figure 6.** Images observed by the visible (VIS) cameras between 26–30 December 2018 of the Etna eruption. The ECV camera is installed in Catania at Catania (CUAD) station south of Mt Etna, while EBVH in Bronte on the western flank of the volcano.

VPTH results, either obtained by the analysis of the satellite images of 24 December and by the ground-based camera images from 26 to 30 December, which were used as input for the VPR procedure to retrieve ash and $SO_2$ parameters, are reported in Table 1.

**Table 1.** VPTH from satellite and ground-based VIS camera images from 24 to 30 December 2018 used for the VPR ash and $SO_2$ retrievals.

| DateDecember 2018 | VPTH [km] |
| --- | --- |
| 24 | 8.0 |
| 26 | 4.0 |
| 27 | 4.5 |
| 28 | 5.5 |
| 29 | 4.5 |
| 30 | 4.5 |

### 3.4. Volcanic Ash and SO$_2$

The ash mass, AOD, R$_e$, and SO$_2$ mass were obtained by processing the SEVIRI mages every 15 min with the VPR procedure. Figure 7 shows the time series of SO$_2$ total mass (M$_s$, upper plot), ash total mass (M$_a$, middle plot), and ash R$_e$ (lower plot). The error bars are set to 40% of the values retrieved [53]. As in Figures 3 and 4, the green area indicates the interval of time in which a wide meteorological cloud system was over the Mediterranean area, preventing any reliable retrievals. The graph shows that the SO$_2$ mass is generally higher than the ash mass, except on 24 December and partially on 29 December. SO$_2$ retrievals show different peaks greater than 15 kt from 27 to 29 December, while the maximum ash value is obtained 29 December with about 20 kt. The mean R$_e$ is about 4 μm on 24 December and about 7 μm for the rest of the period. This is justified by the presence of the climax of the eruptive activity only on 24 December. Note that the VPR procedure corrects for the effect of the volcanic ash on the channels used for the SO$_2$ retrievals. Uncertainty on the ash optical properties (derived from the ash refractive index) could lead to significant uncertainty also on SO$_2$ retrievals.

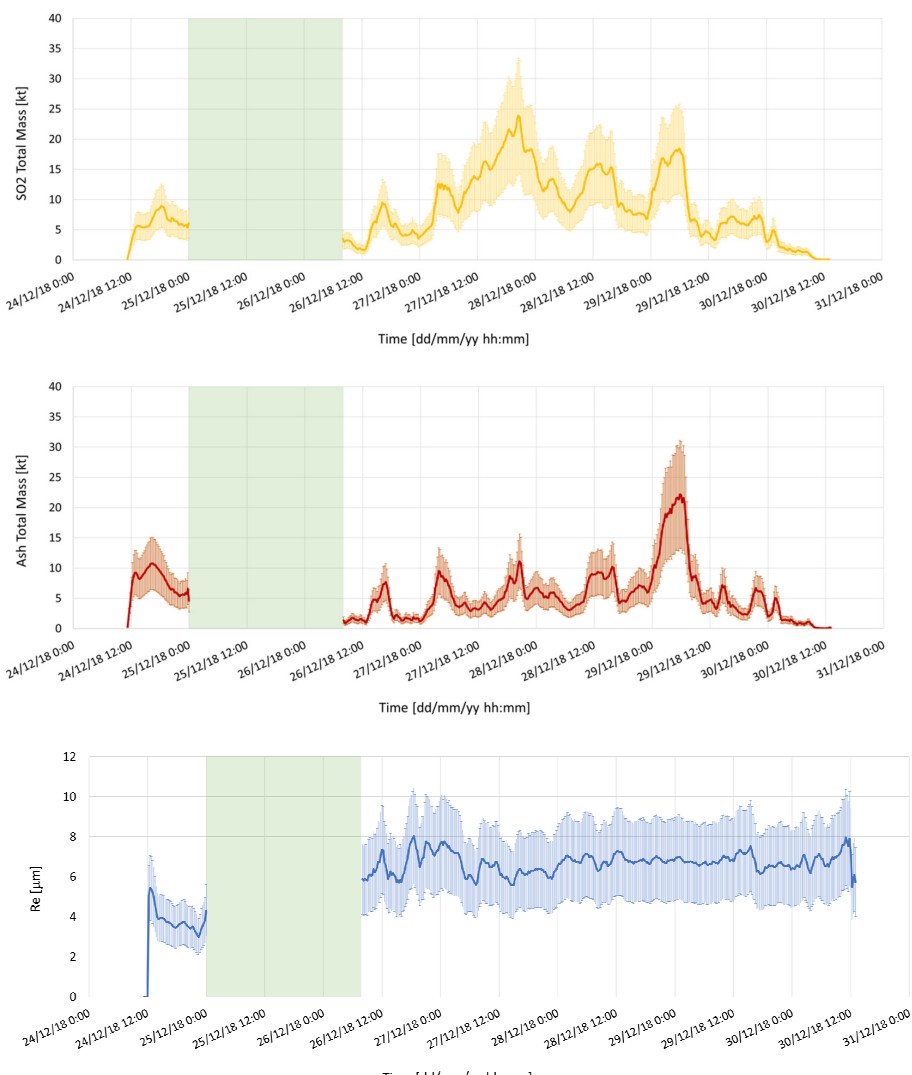

**Figure 7.** Time series of SO$_2$ total mass (upper plot), ash total mass (middle plot), and ash mean Re (lower plot) obtained from all the SEVIRI images with a 15-min time step. In green, the time interval for which a meteorological cloud system was over the Mediterranean region. The vertical bars represent the respective SEVIRI retrieval uncertainties.

### 3.5. Volcanic Ash and $SO_2$ Fluxes

The ash and $SO_2$ fluxes were computed from each 15-min SEVIRI image considering only pixels at a distance of 30 +/− 1.5 km from the summit craters and neglecting those very close to the crater with large opacity, which may lead to a large uncertainty in the retrievals. The plume-wind speed values were obtained from the ARPA atmospheric vertical profiles [52] for the entire period in 6-h steps (00, 06, 12, 18) and interpolated at VPTH values (Table 1). Then, the flux was computed using the following equation:

$$F(t) = l * v(t) * \sum_{i=1}^{n} M_i \,,\tag{1}$$

where l is the transect width (in this case 3000 m), v is the wind speed (m/s), and $M_i$ is the $SO_2$ columnar abundance (kg/s) for a certain pixel. By means of the wind speed, the flux at 30 km is then reported to 15 km, which is immediately comparable with the FLAME measurements (see Section 4.3). Figure 8 shows the $SO_2$ flux over the eruptive period. The $SO_2$ emission rates were highly variable with peaks that exceed 600 kg/s on 24, 27, and 28 December and a mean value over the whole period of 185 kg/s.

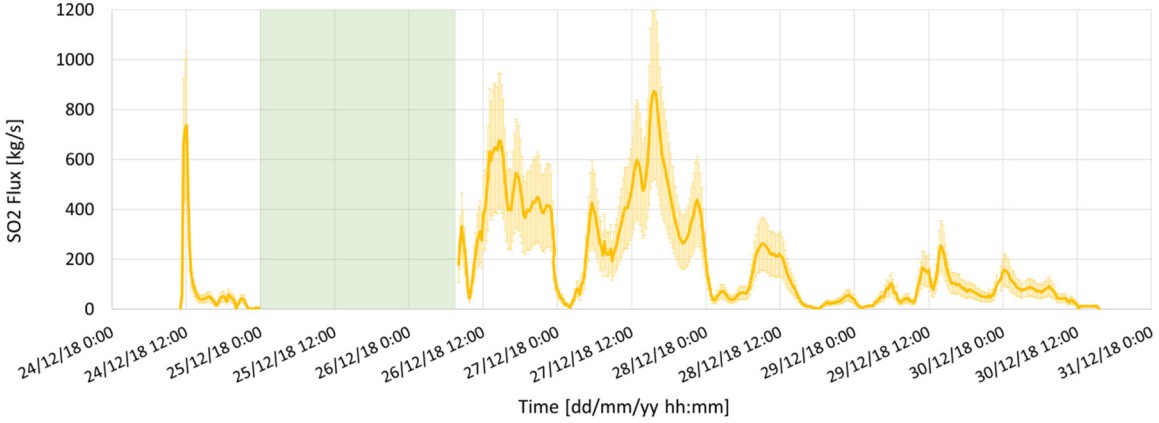

**Figure 8.** $SO_2$ flux time series obtained from 15-min SEVIRI images. In green, the time interval that indicates the presence of a meteorological cloud system over the Mediterranean. The yellow vertical bars represent the SEVIRI $SO_2$ flux retrieval uncertainties.

Similarly, to $SO_2$, the ash flux was computed as well. The ash emission rates have generally lower values than $SO_2$ except for those on 24 December, when there was a huge peak above 1500 kg/s. The ash flux mean value over the whole period is 67 kg/s. By integrating the fluxes over the whole period 24–30 December, it is possible to retrieve the total mass emitted by the volcano, which results in about 100 and 35 kt for $SO_2$ and ash, respectively.

## 4. Validation

In this section, the SEVIRI proximal and distal eruption parameters have been compared with those obtained from the ground using the VIS cameras and FLAME networks and from the satellite by exploiting the measurements of the MODIS satellite instruments on board the Terra and Aqua polar satellites.

### 4.1. VPTH Using Ground-Based VIS Camera

Section 3.3 introduced the ground-based VIS cameras used for the VPTH validation. Figure 9 shows the VPTH comparison between the filtered data obtained with the ECV camera installed at CUAD in Catania and SEVIRI data collected on 24 December. As the figure shows, the two curves are in close agreement until 13:30 to temporally decouple, starting from 13:45 with the SEVIRI retrievals rising higher than the camera estimations. Before 13:45, our SEVIRI estimations are also consistent with those obtained from Calvari et al. [45], which uses the ECV VIS camera and the approach developed by Scollo

et al. [11]. An agreement is also found between the maximum height of the volcanic cloud (VCTH) obtained from the VIIRS image collected on 24 December at 12:46 (8.8 +/− 0.2 km) [45] and the VCTH obtained from SEVIRI approximately at the same time (7.9 +/− 0.3 km at 12:45). Differently, the high VPTH's values (more than 13 km at about 11:30) estimated in [45] by the uncalibrated wide-angle VIS camera (ECVH), located at CUAD in Catania next to the ECV (see Figure 1), are not supported by both our ground-based and satellite retrievals.

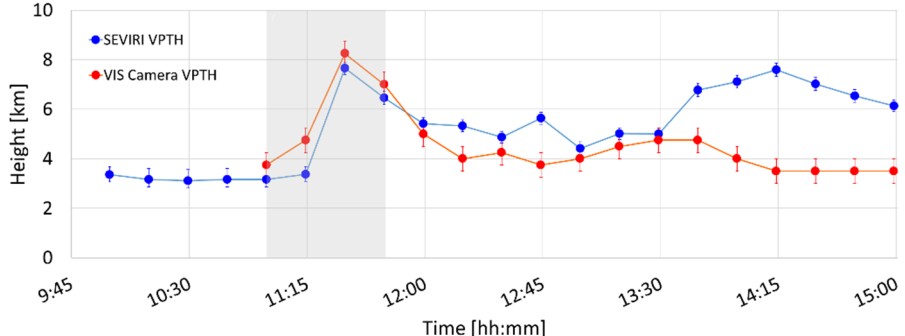

**Figure 9.** VPTH obtained from SEVIRI (blue line) and ECV VIS camera (red line) on 24 December 2018. The blue and red vertical bars represent the SEVIRI and VIS camera VPTH retrieval uncertainties, respectively.

In order to deepen the discrepancy between the SEVIRI and ECV retrievals after 13:30, the images of the ECVH camera were considered. In Figure 10, the image on the upper left shows the ECVH frame collected on 24 December at 11:30, while in the upper right, the same image is displayed together with the image collected by the ECV camera at the same time (red box) and the SEVIRI view (blue dashed lines, 9 × 9 pixels centred on Etna craters). Below, the time frames of Etna eruption observed by the ECVH camera between 11:15 and 15:00 every 15 min are shown. From about 12:00, the images show the gradual formation of a wide orographic lenticular cloud over the volcanic one, with features similar to the "Contessa" cloud that often covers the top of Etna, but bigger and located far from the craters. This cloud partially affected the SEVIRI view from the 13:45 onwards. As a result of that, from this time, the SEVIRI VPTH is not the height of the volcanic column but instead the height of this orographic cloud.

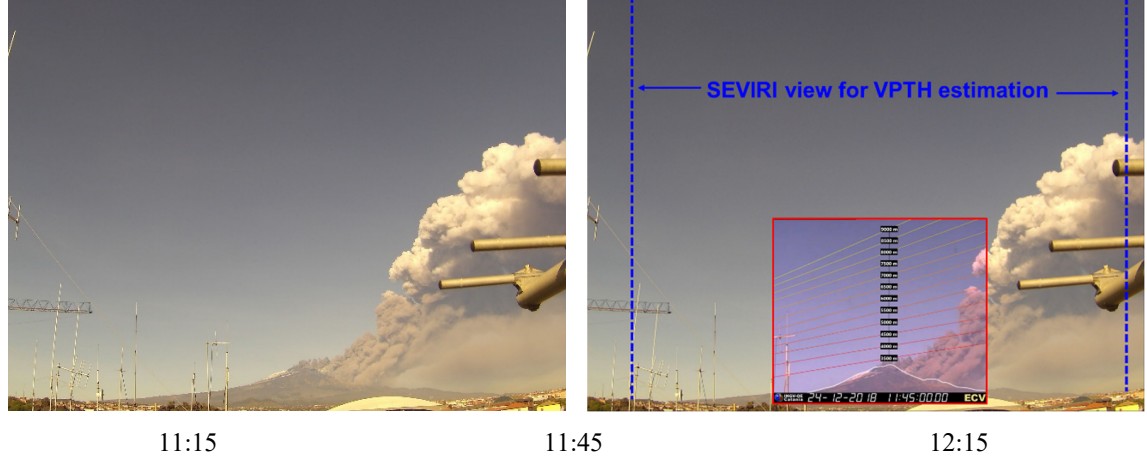

**Figure 10.** *Cont.*

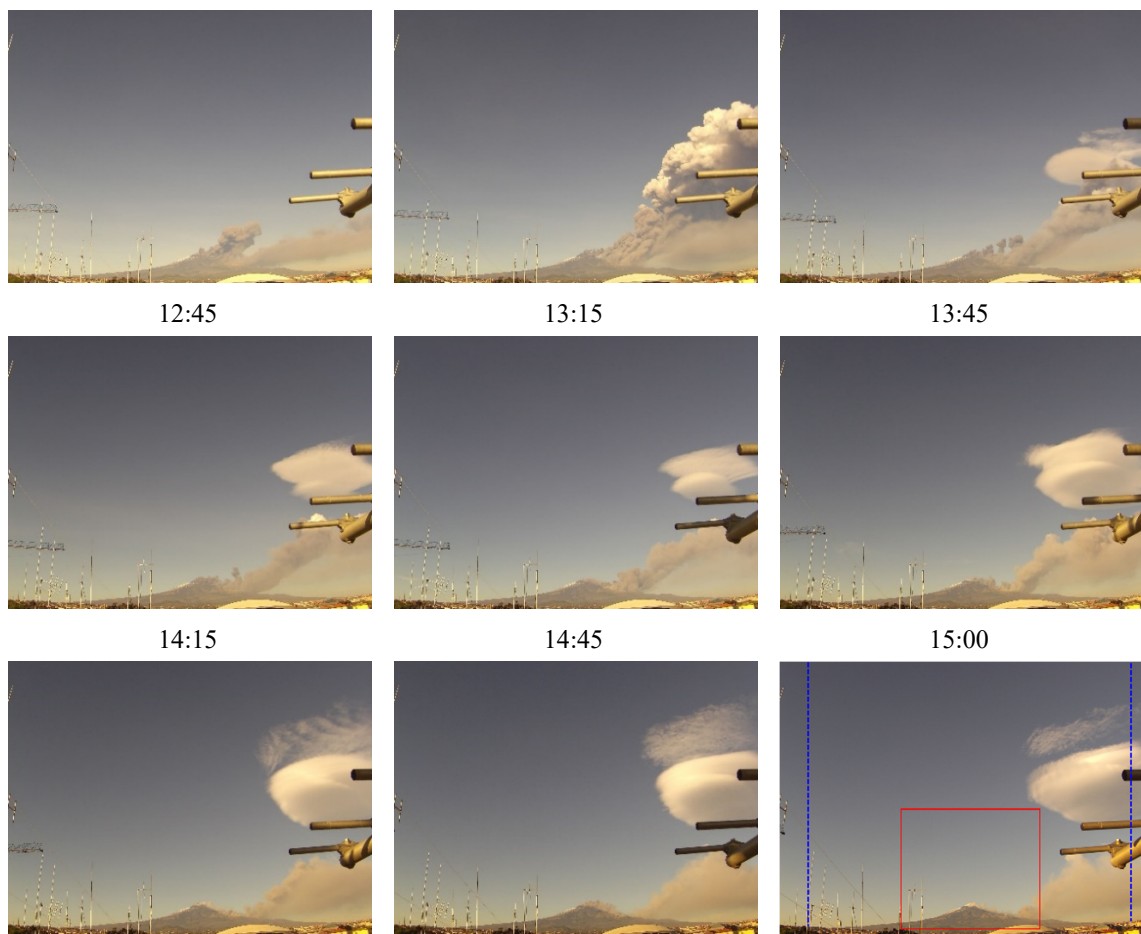

**Figure 10.** Upper left: ECVH camera frame collected at 11:45 on 24 December 2018. Upper right plate: same ECVH frame with the superimposition of the simultaneous ECV camera image and the SEVIRI VCTH view. Below, time series of the ECVH frames collected from 11:15 to 13:00, showing the formation of an orographic lenticular cloud over the volcanic plume. In the 15:00 time frame, the VIS camera and SEVIRI views (red box and blue dashed lines) are displayed for visual comparison.

### 4.2. Ash and $SO_2$ Mass Using MODIS Data

MODIS, aboard the NASA-Terra/Aqua polar orbit satellites, is a multispectral instrument with 36 channels from VIS to TIR, a spatial resolution of 1 km, and a revisit time of 1–2 days [54]. Table 2 summarizes the MODIS images used for the cross-comparison.

**Table 2.** MODIS images considered for the SEVIRI ash and $SO_2$ cross-comparisons.

| December 2019 | MODIS |
|---|---|
| 24 | 11:50 (Aqua) |
| 26 | 11:35 (Aqua) |
| 27 | 12:20 (Aqua) |
| 28 | 11:25 (Aqua) |
| 29 | 12:05 (Aqua) |
| 30 | 09:35 (Terra) |

$SO_2$ and ash mass retrievals from MODIS were obtained with the same VPR procedure applied to SEVIRI data. Figure 11 shows the cross-comparison between the SEVIRI and MODIS results for $SO_2$ (upper plot) and ash (lower plot). This figure shows that the MODIS and SEVIRI retrievals are in very good agreement, as all estimates were within measurement errors.

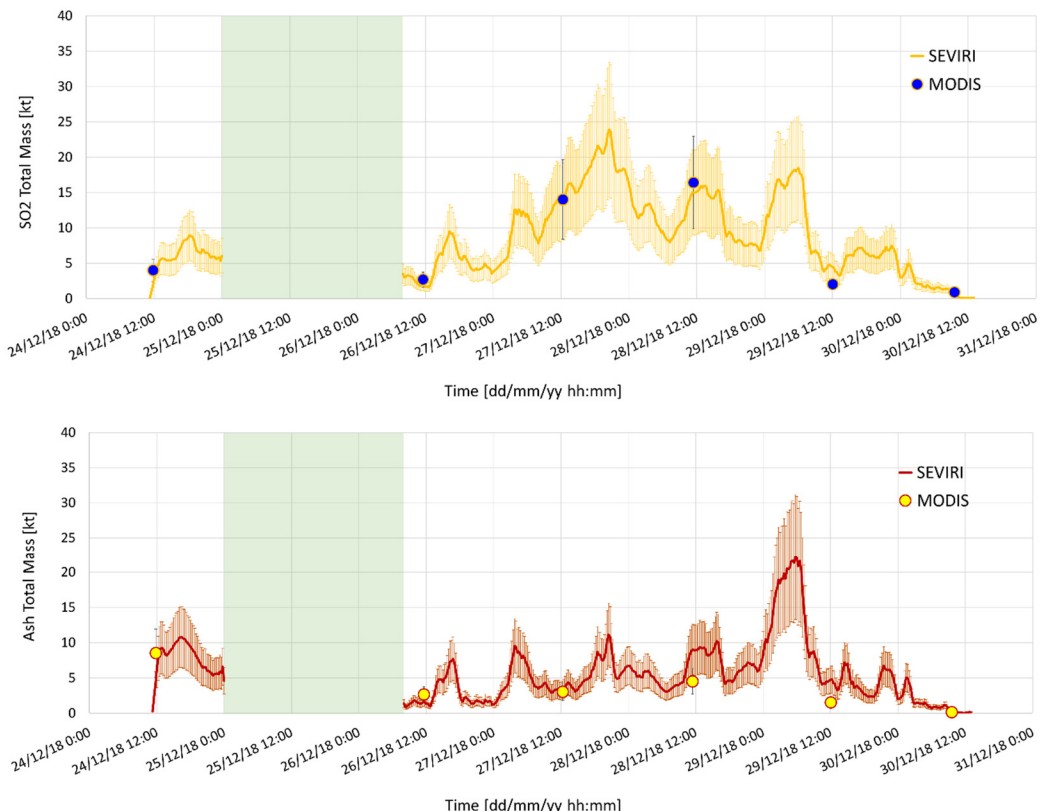

**Figure 11.** SO$_2$ (upper plot) and ash (lower plot) total mass retrievals obtained from SEVIRI and MODIS. The vertical bars represent the SEVIRI retrieval uncertainties.

### 4.3. SO$_2$ Flux Using MODIS and Ground-Based FLAME Data

The SO$_2$ flux cross-comparison was carried out by exploiting the MODIS and the ground-based DOASFLAME measurements.

The MODIS SO$_2$ flux was computed in a very similar manner to that used for SEVIRI, with the only difference that only one transect (at 30 km) for each image was used for SEVIRI; instead, for MODIS each plume-perpendicular axis transect was placed at an increasing distance from the crater. Finally, the MODIS flux is reported to 15 km to be immediately comparable with FLAME and SEVIRI. The application of this procedure to one single image leads to the flux reconstruction over a larger time range [34,35]. The daylight bulk sulfur dioxide (SO$_2$) flux from the summit craters of Mt. Etna was measured also by the ground-based Ultraviolet scanning DOAS spectrometer Network FLAME (FLux Automatic MEasurements). The network consists of 10 ultraviolet scanning spectrometer stations spaced approximately 7 km apart and installed at an altitude of approximately 900 m asl on the flanks of Etna. Each station scans the sky for almost 9 h, intersecting the plume at a mean distance of approximately 14 km from the summit craters and acquires a complete plume-scan of approximately 5 min. Open-path ultraviolet spectra are inverted on site, applying the DOAS technique and using a modeled clear-sky spectrum [55]. Inverted data are in real time transmitted to the INGV, Osservatorio Etneo, where SO$_2$ emission rates are computed. Uncertainty in flux range between −22 and +36%. Plume height is essential to constrain the measurements for the fixed scanning mode of SO$_2$ measurements, and an accurate estimate of height can be obtained by triangulation using a pair of scanners. Nevertheless, this method might be challenging, as it requires a stable prevailing wind coupled with a high spatial number of scanner station in the network. Therefore, given the main monitoring purpose, we adopt a new strategy in which plume height is inferred using an empirical relationship between plume height and wind-plume transport speed [55]. This approach considers a maximum plume altitude of 5500 m; thus, the altitude estimated might be affected by uncertainty in

the case of the paroxysmal eruptive episode. Therefore, if the $SO_2$ flux computed using such a strategy is acceptable for real-time monitoring and volcanology purposes [45], a post-processing using an accurate height is crucial for specific objectives, as comparison among ground and satellite retrievals. In this work, the $SO_2$ heights and wind speed were set with the same values used for the SEVIRI and MODIS flux retrievals. Figure 12 shows the comparison between the SEVIRI, MODIS, and FLAME $SO_2$ fluxes. Except for on 24 December, the SEVIRI and MODIS $SO_2$ fluxes are in good agreement; conversely, for FLAME, a greater discrepancy was found. In order to inspect these differences, the data were analyzed, considering them daily (Figure 13).

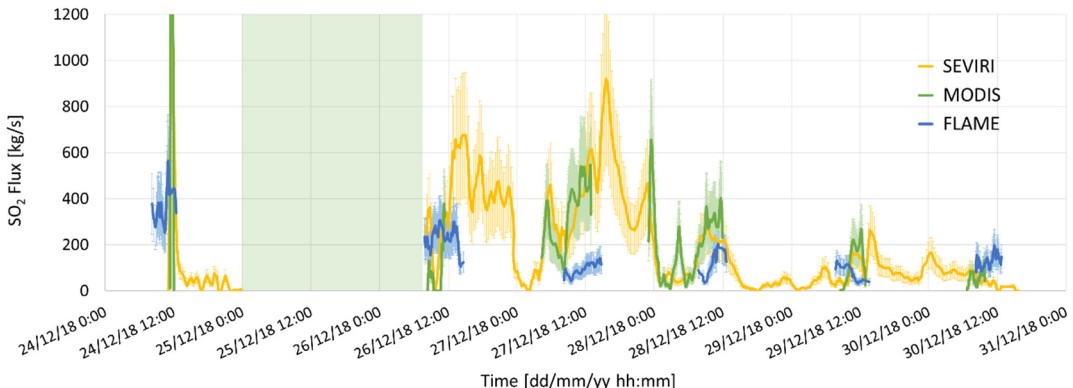

**Figure 12.** $SO_2$ flux obtained from SEVIRI (yellow), MODIS (green), and FLAME (blue). The yellow vertical bars represent the SEVIRI retrieval uncertainties.

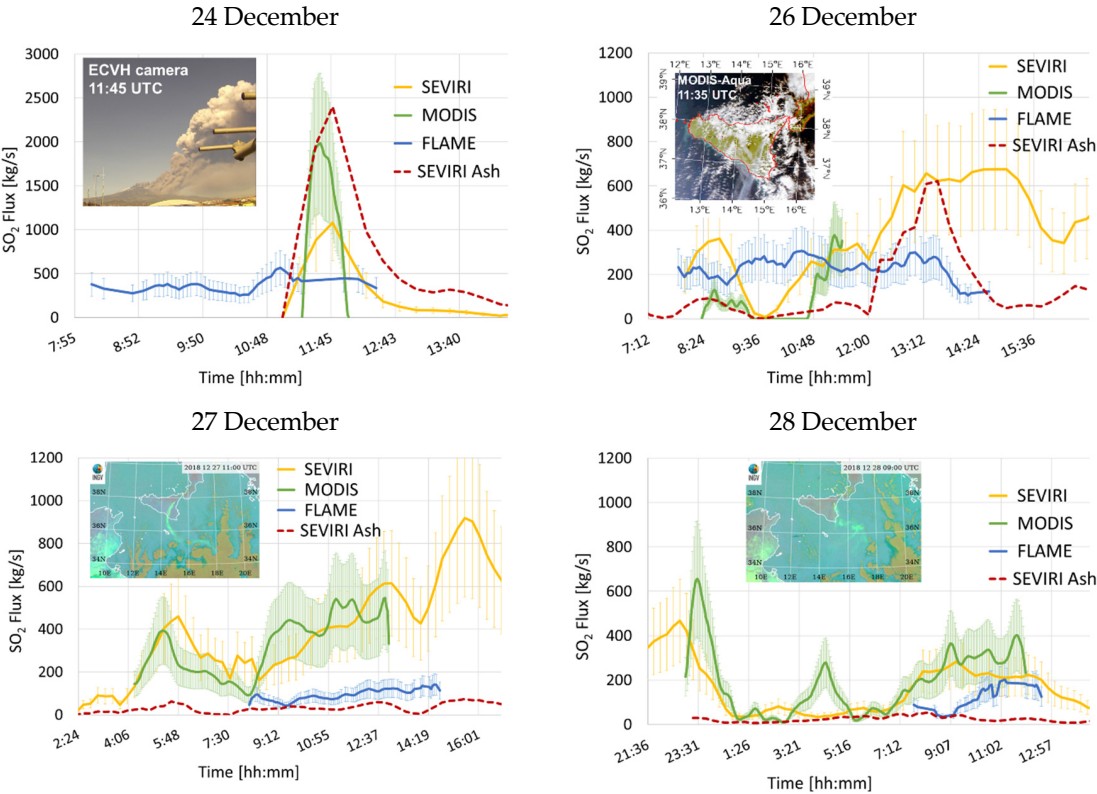

**Figure 13.** Detailed comparison of $SO_2$ fluxes for the 24, 26, 27, and 28 December observed by SEVIRI, MODIS, and FLAME. The dashed maroon line indicates the ash flux obtained from SEVIRI measurements.

The 24 December plot (upper left plate of Figure 13) shows that only the FLAME network retrieved an $SO_2$ flux before the beginning of the paroxysm that occurred at 11:15. The detection failure of SEVIRI

and MODIS is due to the ground-based plume, which yields a too-small thermal contrast between the top of the volcanic plume and the underlying ground surface, thus making the satellite $SO_2$ retrieval impossible. Moreover, the discrepancy observed during the paroxysm is also caused by high ash load in the plume, as highlighted by the ash flux curve of Figure 13 (see the dashed maroon line) and by the ECVH image collected at 11:45 (upper left plate in the plot). The ash presence has an impact on both the $SO_2$ retrieval from satellite (the greater the uncertainty in the ash optical properties, the greater the errors of the ash correction for the $SO_2$ retrievals [56]) and FLAME (by obscuring the signal).

The low $SO_2$ fluxes values obtained from the satellite measurements around 9:00 of the 26 December (upper right plate of Figure 13) are due to the presence of meteorological clouds in the Etnean region, as shown in the MODIS image collected at 11:35 (upper left graph), which partially obscured the volcanic cloud signal to the satellite. The lower FLAME $SO_2$ fluxes, compared with SEVIRI/MODIS, after 12:00 is due to the presence of the high amount of ash in the volcanic plume, as indicated by the dashed maroon line in the plot.

Conversely, the lower FLAME $SO_2$ fluxes compared with SEVIRI/MODIS retrievals observed in the case of 27 and 28 December (Figure 13) are not related to the ash in the plume (the dashed maroon line indicates extremely low ash values), but instead to a gap of data of the FLAME station in Nicolosi (see the ENIC station in Figure 1) due to a technical problem. As a matter of fact, due to the principal direction of dispersion of the volcanic plume toward the south (south-west and south-east for 27 and 28 December, respectively, see SEVIRI RGB images in the plots), the FLAME network could only partially observe the $SO_2$ plume over these days, leading to a significant flux underestimation compared with the space-based observations.

## 5. Conclusions

In this work, the ability of the SEVIRI sensor, on board the MSG geostationary satellite, to follow the whole evolution of a volcanic activity from the source to the atmosphere in near real time is presented. The accurate estimation of the proximal and distal parameters, characterizing the volcanic activity in near real time, is the key issue for the mitigation of the effect of volcanic eruption on local population and airspace. The speed and accuracy at which this information is given is crucial for both the national civil protections (to organize the evacuation of the population when needed) and the international aviation organizations (to communicate the presence of a volcanic cloud in the atmosphere, which allows aircraft to avoid it).

The measurements collected from SEVIRI are used for the characterization of the Etna 24–30 December 2018 eruption. The estimations are realized in the whole period, with a time frequency from 5 to 15 min, with the exception of the period from 25 (at 00:00) to 26 (at 08:00) December, in which a wide meteorological cloud system over the Mediterranean region partially shielded the eruption.

The hot spot analysis indicates the onset of the eruption on 24 December at 11:15. The TADR estimation on 24 December shows an abrupt increase with peak of 8.3 $m^3$/s and a subsequent exponential decrease, indicating the occurrence of a lava flow from the eruptive fracture toward Valle del Bove. The TADR integration gives a total volume of lava emitted of about 0.5 $Mm^3$.

The maximum column height is estimated to be about 8 km asl on 24 December, while for the rest of the period, it ranges from 4 to 5.5 km.

The distal monitoring indicates that the volcanic cloud flowed generally toward the south/south-east and that ash and $SO_2$ are totally collocated. The $SO_2$ burden results were generally higher than ash, and the total amounts were about 100 and 35 kt, respectively. The mean $SO_2$ flux was estimated to be 185 kg/s, with peaks of more than 600 kg/s; instead, the mean $R_e$ was lower on 24 December (4 µm) compared to the rest of the period (7 µm) due to the eruptive climax only on that day.

All the SEVIRI products were compared with those obtained from the ground-based (VIS cameras and FLAME network) and polar satellite (MODIS) measurements. The VPTH, ash, and $SO_2$ parameters cross-comparison indicates a good agreement between SEVIRI, camera VIS, and MODIS retrievals

with estimations that all lie within measurement errors. The VPTH discrepancy after 13:00 on 24 December found between the SEVIRI and CAMERA VIS is due to the gradual growth of a wide orographic lenticular cloud over the plume. The SEVIRI and FLAME $SO_2$ flux retrievals show significant discrepancies due to the presence of volcanic ash that affect both the retrievals and a gap of FLAME data between 27 and 29 December. The validation process will be further expanded through the use of other satellite systems and procedures in a work currently in preparation.

The results achieved in this work prove the ability of remote sensing systems to characterize eruptive events in near real time, offering a powerful tool to mitigate volcanic risk on both local population and airspace.

**Author Contributions:** Conceptualization and work coordination, S.C.; methodology, S.C., L.G.; satellite data processing, S.C., L.G., D.S., M.M., M.S., V.L., L.M.; ground based data processing, G.S., S.S., M.P., T.C.; data analysis and cross-comparison, all; writing—original draft preparation, all. All authors have read and agreed to the published version of the manuscript.

**Funding:** This research was supported by the ESA project VISTA (Volcanic monItoring using SenTinel sensors by an integrated Approach), grant number 4000128399/19/I-DT, by the Italian Ministry of Education, University, and Research, project FISR2017-SOIR (Decreto n. 3455 del 4/12/2017) and by the Italian MIUR project Premiale Ash-RESILIENCE and by the EU e-shape project, grant agreement: 820852.

**Acknowledgments:** The authors would like to thank EUMETSAT and NASA for providing SEVIRI and MODIS data, and all the technicians of INGV-OE who are involved in the maintenance of the camera and $SO_2$ monitoring networks.

**Conflicts of Interest:** The authors declare no conflict of interest.

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
