# Peer review of "Near Real-Time Monitoring of the Christmas 2018 Etna Eruption Using SEVIRI and Products Validation"

_remotesensing, doi:10.3390/rs12081336_

Round 1

Reviewer 1 Report

I have received the review request of “Near real time monitoring of the Christmas 2018 Etna 3 eruption using SEVIRI and products validation” and would like to thank you for giving me this opportunity to review this interesting article. The authors have documented the 24 December 2018 Etna Volcano Eruption event. Their idea of using the near-real-time monitoring technology to record volcano eruption is interesting. Not only have they adopted various tools, but also their data are clearly documented. However, my first and major concern is that the data need to be better organized. Therefore, I would personally suggest that the authors can perhaps revise the structure of this article in order to highlight scientific contributions thereof instead of being a documentary data. Therefore, my overall recommendation for this paper is to reconsider it after major revision.

My suggestions for the authors are as follows:

Structure

  1. By taking Section 5 as an example, the authors have adopted various tools to valid the SERVRI results in three subsections, such as VPTH using VIS camera and Ash and SO2 mass using MODIS data. However, instead of comparing the data of different tools, the authors can perhaps highlight the message that they wish to deliberate to the readers.
  2. In professional periodicals, it is not often to introduce the article structure in Introduction (Line 64 to 67). It is suggested that the authors can perhaps reconsider the necessity thereof and make adjustments accordingly.
  3. The authors are suggested to check the position of tables and figures. For example, Figure 2 is not nearby the text that describes Figure 2; Figure 7 is not nearby the text that describes Figure 7; Line 212 seems to be figure, but there is no figure descriptions.
  4. With respect to author contributions, all of the authors’ names are written in abbreviations and without their organization. It is perhaps not necessary to specify the organization, but writing their names in abbreviations is not recommended.

Data Presentation

  1. Section 5.3 (Figure 13 and 14): some discrepancies are found in curves in this section. The authors are suggested to explain the discrepancy and similarity of the SERVIRI, MODIS and FLAME curves as well as their respective limitations.
  2. Figure 14: The SEVIRI Ash legend on 26 December is missing.

Data Application

The authors’ purpose is to use near-real-time monitoring technology to characterize eruptive events and to perhaps offer a powerful disaster mitigation solution (Line 374-376). However, from the perspective of disaster prevention and monitoring, it will be important and interesting to know the disaster response time of SERVRI system. That is, how long does it take from data acquisition to data analysis to giving an alert message.

English writing

This paper is indeed well presented in terms of English writing. However, it is suggested that the authors can perhaps review the paper again for minor English problems. Examples are as follows:

  1. Line 14, 25 and so on: It is suggested that the authors can unify the writing of date. For example, “the 24th of December” or “24 December” to make the English writing more consistent.
  2. Line 15: a strong shallow earthquake-s (remove “s”?)
  3. Line 24: from 4 to 5.5 km (4 cm or km? if no hyphen is used between the numbers, please specify the unit thereof)
  4. Line 43: may cause-s (remove “s”?)
  5. Line 145: from 15 to 5 minutes (“15 seconds to 5 minutes” or “5 minutes to 15 minutes”?)
  6. Line 165: is showed (shown?)
  7. Line 261: Section3.3 (Section 3.3?)
  8. Please check the punctuation marks. Ex. Lines 208, 214, 271 and so on.

Author Response

Authors would like to thank the anonymous referee for comments and suggestions that helped improving the clarity of the whole paper. The new version has been widely revised as requested, in particularly by improving the introduction, retrievals methodology description, results description and conclusions. Also different figures have been improved.

Best Regards

Stefano Corradini

Reviewer 2 Report

Authors presented a near real time monitoring approach of the Etna eruption from December 2018 and compared with other ancillary data. Authors used combination of geostationary and polar-orbit satellite data and ground based information obtained via SEVIRI, MODIS instruments and VIS camera, respectively. Technologies and results presented allow to better understand volcanic processes and mitigate volcanic hazard and risk in local and regional scales.

            Manuscript is interesting and research performed useful and of the great importance for natural scientists and human population in general. However, some editorial corrections have to be done. The structure of the manuscript is not ordered. Please reorder the whole manuscript according to the publishable standards.

Selected remarks to the manuscript:

  1. In my opinion “…Christmas Etna eruption…” is not a correct title for the scientific paper. This is kind of religious characterization or sounds like a Christmas fireworks…
  2. Please add geological/geomorphological setting and present a localization of the test-site. (We do not even know where Valle del Bove is located?( Lines – 17, 76)
  3. Please add methodology section. Please add also clear measurements’ errors characterization and consistent procedures’ limitations.
  4. e.g. lines 153-162 – should be moved to the Methodology section.
  5. Please add consistent data characterization section.
  6. Line 62 – “on board”, please correct.
  7. Please unify date format in the whole manuscript.
  8. Figure 2 is not placed correctly, please reorder it.
  9. Sections, subsections and their titles should be enumerated correctly and named with one code and scheme, please correct: e.g. vide sections (with the order of authors’ numeration): 3, 4, then 3.1, 3.2, 3.3., 3.4 and 5. It is messy edition, or better no edition so far.
  10. Figure 7 is double, please also order the caption.
  11. Table 2 should be moved to data section
  12. Line 261 – section 3.3?, please correct.
  13. Figure 14 - small images are not informative.
  14. Lines 375-376 “…offering a powerful tool to mitigate volcanic risk on both local population and airspace” this part of the sentence is not a conclusion coming from this paper. Maybe authors should add discussion section?
  15. I gave the authors just the examples of editorial errors, please check the whole manuscript to make it readable.

Author Response

Authors would like to thank the anonymous referee for comments and suggestions that helped improving the clarity of the whole paper. The new version has been widely revised as requested, in particularly by improving the introduction, retrievals methodology description, results description and conclusions. Also different figures have been improved.

Best Regards,

Stefano Corradini

Reviewer 3 Report

General Comments

This manuscript addresses a highly relevant topic on how to monitoring using satellite-based data a volcano eruption in NRT. The manuscript contribution will support to improve the volcano monitoring in low dense stations network, supporting to volcano observatories in developing countries. However, I am concerned about the presentation and the methodological formulation proposed. The methodology needs to be improved, some results correspond to methodology, and uncertainty analysis is needed here. Moreover, the results need to be extensive explained. Some figures don’t support the quality of the contribution. I encourage the authors to improve the manuscript that “could be” a good contribution to Remote Sensing.

Specific Comments:

Abstract
The scientific question is not well introduced. According to my understanding, the manuscript introduces a study case. However, the scientific question fails to be introduced.

Introduction
The introduction should be improved urgently. Please, review the state of the art that support your study. The authors need to justify the physical parameters (TADR, VPTH, VCTH, ESD) estimated/measured that could be useful to volcano monitoring. I suggest a deep review here. Please be more explicit in the scientific question. The lack of an explicit scientific question is the main problem in the manuscript.
Section 2
Please, justify the implementation on Etna Volcano. I understand that some instruments are available here. However, the authors should be more specific of the volcanic setting that allow carry out the experiment reported in the manuscript.
Section 3
The methodology is very poor. Additional methodological information is presented in section 4, please move here. I encourage the authors to improve the text. It’s mandatory to be suitable to publish. Please increase the description of the methodology more than use references. Moreover, please clarify if RGB used here correspond to “true RGB” or correspond to the “composite RGB” inline 116. Line 117 fails to explain that the eruption corresponds to an SO2-rich cloud.
Results
The results section is hard to read if I don’t have an explicit research question or scientific hypothesis. The authors should declare their hypothesis urgently.
Please rewrite lines 138-140.
Figure 4 needs to be improved
Line 310-313 correspond to methodology
Section 5
Given all of the above methodological deficiencies, the discussion section does not provide any clear indication of anything. However, these issues are susceptibles to be improved rewriting the current manuscript.

Author Response

(The authors gave the same response as above.)

Round 2

Reviewer 3 Report

Dear authors

The manuscript shows a significant improvement. I want to express my sincere congratulations to the research team. However, several typos must be corrected shortly.

Lin 121: Please, add a reference. Moreover, explain better the relationship between DR and DT

Lin 132: please, add space

Lin 148: Please, try to connect better your idea

Lin 161: The authors need to improve section delimitation.

Lin 155-161: I don't understand why the text is in bold.

Lin 244: I suggest merging figures 6, 10 and 11 in just one. 

Lin 286: Validation is very light. It is difficult to evaluate the performance of the model using only 5 SEVERIS / MODIS comparison points. I suggest to extend the time series at least 15 days before and 15 days later if there is a possibility of a clear sky. Otherwise, the authors will need to extend more the series. I request to the author that compute the uncertainties derived from the model.

Lin 373: figure inserted wrongly

Lin 384: figure inserted wrongly

Lin 390: figure inserted wrongly

Lin 397: figure inserted wrongly, the authors need to be careful.

Lin 409: The authors need to evaluate the quality of their validation. I recommend reviewing their conclusions.

Additional comment: I encourage the author to share their codes and some training test for the coming review round.
